# A New Grafting Method for Watermelon to Inhibit Rootstock Regrowth and Enhance Scion Growth

Changjin Liu [1], Weiguo Lin [2], Chongran Feng [1], Xiangshuai Wu [1], Xiaohu Fu [2], Mu Xiong [1], Zhilong Bie [1] and Yuan Huang [1,3],*

[1] College of Horticulture and Forestry Sciences/Key Laboratory of Horticultural Plant Biology, Ministry of Education, Huazhong Agricultural University, Wuhan 430070, China; LiuCJ@webmail.hzau.edu.cn (C.L.); fengcr@hotmail.com (C.F.); 2020305120109@webmail.hzau.edu.cn (X.W.); Muxiong@webmail.hzau.edu.cn (M.X.); biezl@mail.hzau.edu.cn (Z.B.)

[2] College of Engineering, Huazhong Agricultural University, Wuhan 430070, China; linweiguo@mail.hzau.edu.cn (W.L.); fuxiaohu6970@gmail.com (X.F.)

[3] Guangdong Laboratory of Lingnan Modern Agriculture, Genome Analysis Laboratory of the Ministry of Agriculture, Agricultural Genomics Institute at Shenzhen, Chinese Academy of Agricultural Sciences, Shenzhen 518000, China

\* Correspondence: huangyuan@mail.hzau.edu.cn; Tel.: +86-27-8728-2010

**Abstract:** Grafting is a widely used technique in watermelon (*Citrullus lanatus*) production. However, cost of grafted seedlings is generally high as a result of intensive labor inputs for propagation using traditional grafting methods such as the manual removal of rootstock regrowth. This study developed a new grafting tool to physically remove the epidermis of pumpkin (*C. maxima* × *C. moschata*) and bottle gourd (*Lagenaria siceraria*) rootstock cotyledon base during grafting; we called this a new grafting method. Compared with the traditional grafting, the new grafting method significantly decreased the pumpkin rootstock regrowth rate from 100% to 8% in hole insertion and 2% in one cotyledon grafting, respectively. These attenuated rates for bottle gourd rootstock regrowth were 23% and 9% in hole insertion and one cotyledon grafting, respectively. The scion dry weights of new hole insertion and one cotyledon grafting were increased by 78% and 74% when pumpkin was used as rootstock as compared with traditional grafting without regrown rootstock removal, while the respective values were 33% and 17% in bottle gourd rootstock grafted plants. In addition, the time used for the new hole insertion grafting method to physically remove the epidermis of pumpkin rootstock cotyledon base was significantly shorter than the time required to remove the rootstock regrowth manually three times in the traditional grafting (4.2 s/plant vs. 9.3 s/plant). Similar results were also observed in the new one cotyledon grafting (4.2 s/plant vs. 8.8 s/plant). Taken together, this study presents a new method in watermelon grafting to reduce rootstock regrowth, therefore benefiting both scion growth and plant management, thus the development of this new method is clearly useful in watermelon production.

**Keywords:** *Citrullus lanatus*; grafting tool; rootstock regrowth; scion growth; vegetable grafting

## 1. Introduction

Grafting plants permits us to select and combine different rootstocks and scions for disease resistance, abiotic stress tolerance, and enhanced yield traits [1,2]. Watermelon is an important cucurbit crop; the world production of watermelon was 101 million tons, and the area harvested was 3.09 million ha hectares in 2019. China is the world's largest producer of watermelon, where the watermelon production was 61 million tons, whereas the planting area was 1.47 million ha in 2019 (http://www.fao.org/faostat/en/#home, accessed on 22 August 2021). Grafting is a commonly used technique in watermelon production. About 40%of watermelon plants are grafted in China, and over 95% of commercial watermelon seedlings are grafted in Japan, Korea, Greece, Israel, and Turkey [3–5]. The main purposes

of watermelon grafting are to solve fusarium wilt and root-knot nematode problems [6,7], to increase nutrient use efficiency [8], tolerance to salinity [9], cold, and drought [10,11], and to increase fruit yield [12]. Huang et al. [12] reported that the productivity of grafted watermelon can be 13%–25% higher than nongrafted plants.

Despite the above advantages of grafting in watermelon production, the cost of grafted watermelon transplants is relatively higher, as it can be up to five times greater than nongrafted plants, with labor representing 48% to 60% of the total cost in a manual grafting operation, which undoubtedly limits the wide use of this technique [5]. Grafting is labor intensive, for instance, management of rootstock regrowth needs to be done manually [13]. Rootstock regrowth can result in graft failure or a decrease in yield by competing with the scion for water and nutrients [5]. The two most common commercial watermelon grafting methods are the hole insertion and the one cotyledon techniques [14]. Rootstock regrowth is unavoidable for the above two grafting methods, since both grafting methods often leave bud meristem tissue at the base of the rootstock cotyledon, resulting in rootstock regrowth occurring after grafting, which is a major problem inhibiting the use of grafted watermelon plants [5]. To solve this problem, some new grafting methods were developed, for example, there is no rootstock regrowth with splice grafting method (both cotyledons removed from the rootstock) because meristem tissue lies below the axillary bud at the base of the cotyledon and is completely removed [5,13,15–17]. In addition, chemical compounds such as fatty alcohol were used to control the rootstock regrowth [18,19].

However, splice grafting often results in a low graft survival rate and poor seeding quality [15,16]. In addition, labor is required for the chemical application, and damage to seedlings can occur [15–19]. Thus, the goal of this study was to establish an alternative approach to control rootstock regrowth. Here, we developed a new grafting method, i.e., the epidermis of rootstock cotyledon base was physically removed by a new tool during grafting. To evaluate the efficiency of this method, rootstock regrowth rate, scion growth, and working time were documented and compared with traditional methods. Compared to traditional methods, significantly lower rootstock regrowth rate and enhanced scion growth were observed in the new method of grafted plants. Meanwhile, this new method was labor-saving since the time used to physically remove the epidermis of rootstock cotyledon base was significantly shorter than the time required by the traditional grafting methods.

## 2. Materials and Methods

### 2.1. Experimental Location and Design

The experiment was conducted in a plant growth room in 2021 at the National Center of Vegetable Improvement in Huazhong Agricultural University, Central China (30°27′ N, 114°20′ E, and altitude 22 m above sea level). The experiments were set up as one-way experiments with three replicates, and each replicate had 25 plants. The factor investigated was grafting method (new grafting method vs. traditional grafting method). The abbreviations and the detailed information of treatments in this study are listed in Table 1.

**Table 1.** Abbreviations and detailed information of treatments in this study.

| Treatment | Detailed Information |
| --- | --- |
| T1 | New "hole insertion grafting" without regrown rootstock removal |
| T2 | New "one cotyledon grafting" without regrown rootstock removal |
| T3 | Traditional "hole insertion grafting" with regrown rootstock removal manually three times |
| S1 | Traditional "one cotyledon grafting" with regrown rootstock removal manually three times |
| S2 | Traditional "hole insertion grafting" without regrown rootstock removal |
| S3 | Traditional "one cotyledon grafting" without regrown rootstock removal |

### 2.2. Plant Material and Cultivation

Watermelon cv. 'Zaojia 8424' (*Citrullus lanatus*, Shanghai Wells Seed Co., Ltd., Shanghai, China) was used as the scion, while interspecific pumpkin hybrid cv. 'Qingyanzhen

No.1′ (*C. maxima* × *C. moschata*, Qingdao Academy of Agricultural Sciences) and bottle gourd cv. 'Jingxinzhen No.1′ (*Lagenaria siceraria*, Jingyan Yinong, Seed Sci-Tech Co., Ltd., Beijing, China) were used as the rootstocks. The scion and the rootstock seeds were sown into 128 and 72-cell trays, respectively, with one seed in one cell filled with seedling substrate (Shandong Shangdao Biotech Co., Ltd., Jinan, China). Mancozeb (1000 times liquid, Hebei Zhongbaolvnong science and technology Co., Ltd., Langfang, China) used in pest control was added in the substrate before seed sowing. During the cultivation, day and night temperatures were 28°C and 18°C, photosynthetic photon flux density was 170 μmol·m$^{-2}$·s$^{-1}$, with 14/10 h photoperiod, and day relative humidity was 65%–85%. The sum of irrigated water for each plug tray was 3.3 L during vegetation. Plants were fertilized with water soluble fertilizer (Product number: 20-10-20 + TE, 1000 times liquid, Hubei Greencare agriculture Co., Ltd., Wuhan, China).

### 2.3. Grafting

One cotyledon grafting was conducted at day 11 after rootstock and scion seeds were sown, about 9 days after the appearance of cotyledons. Hole insertion grafting was conducted at day 11 after rootstock seeds sowing, 7 days after scion seeds sowing and about 5 days after the appearance of scion cotyledons. Grafting was done as described by Hassell et al. [14]. Immediately after grafting, the plants were placed in the healing chambers. The plants were maintained in complete darkness at day 1 and were maintained under low light intensity (80 μmol·m$^{-2}$·s$^{-1}$, 14/10 h photoperiod) from day 2 to day 7. The light intensity was increased to170 μmol·m$^{-2}$·s$^{-1}$ (14/10 h photoperiod) from day 7. The day and the night temperatures were 28°C and 18°C during graft healing. The humidity was kept above 95% during the first 5 days, then decreased to 85% from day 6 to day 10. The plants were removed from the healing chamber at day 10 and were placed in the growth room, following common practice.

### 2.4. New Grafting Tool Development and Usage

The new grafting tool was developed by Huazhong Agricultural University. We already applied for the patent (Application Number: 202110518435.5, China National Intellectual Property Administration). Briefly, one end is composed of a conical file sticking with emery grain, which is convenient to physically remove the epidermis of the rootstock cotyledon base (Figure 1). Another end is the grafting needle, where the purpose is to make a hole in the hole insertion grafting method (Figure 1).

The usage of this new grafting tool is shown in Figure 2a,b, and Supplementary Video S1. For hole insertion grafting, firstly, remove the rootstock true leaf and the growing point, then physically remove the epidermis of the rootstock cotyledon base using the emery grain end. Then, make a hole using the grafting needle end, and finally insert the cut scion. For one cotyledon grafting, firstly, remove the growing point and one cotyledon of rootstock, then physically remove the epidermis of the rootstock cotyledon base using the emery grain end. Finally, hold the rootstock and the scion in place with a grafting clip (Figure 2a,b).

### 2.5. Grafted Survival Measurement

Survival of the grafted plants was assessed at day 14 after grafting. The grafted plants were considered alive and survived if the scion leaves and the rootstock stems were turgid, whereas severely wilted scion leaves and stems of both the scion and the rootstock were considered as graft failure. Survival rate = (Survived number/total number of grafted plants) × 100%.

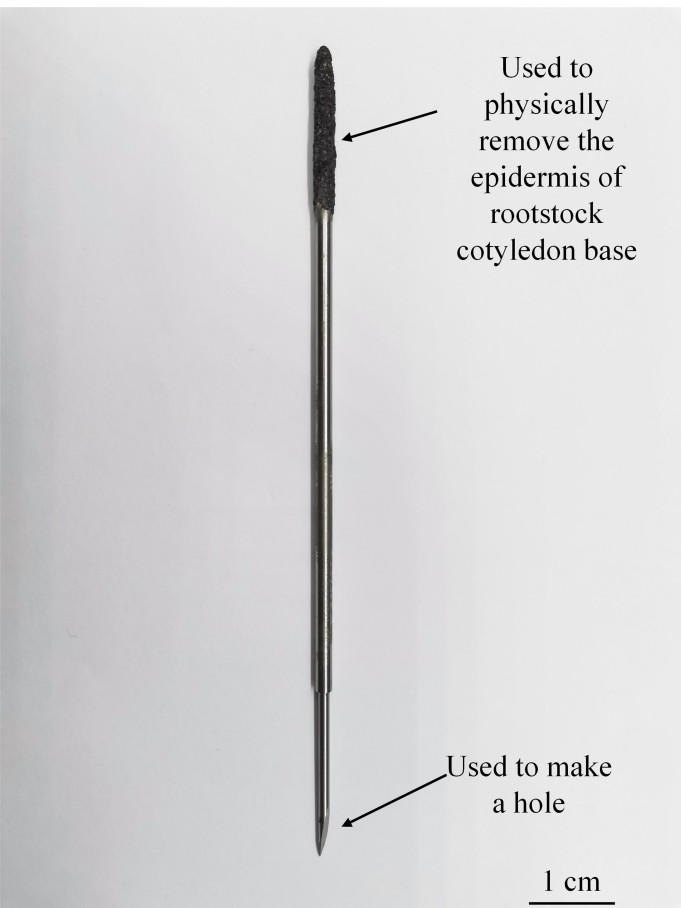

**Figure 1.** New grafting tool used in this study.

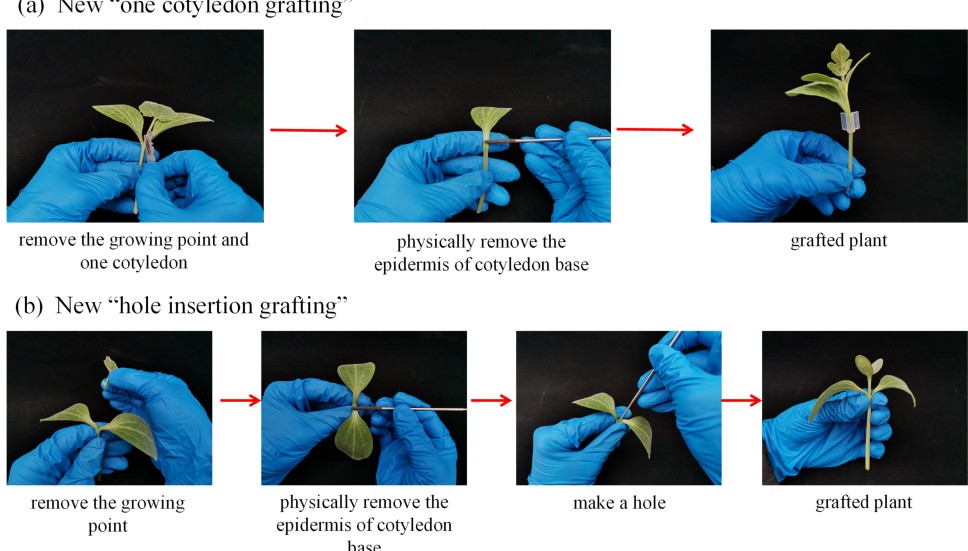

**Figure 2.** Pictures showing how the new "one cotyledon grafting" (**a**) or "hole insertion grafting" (**b**) method is used.

*2.6. Measurement of Rootstock Regrowth Rate*

Rootstock regrowth rate was measured at day 14 after grafting. For this measurement, the regrown rootstock was not removed. Rootstock regrowth rate = (regrown rootstock number/total number of grafted plants) × 100%.

### 2.7. Anatomical Study of Rootstock Cotyledon Base

To observe the rootstock regrowth, histological study of a pumpkin rootstock cotyledon base in the grafted plants was conducted at 0 h, 12 h, 24 h, and 36 h after grafting, using the paraffin section method. The collected samples were placed in 70% FAA for 24 h and then stored in 70% ethanol at 4°C. Paraffin sectioning was performed as described by El-Gazzar et al. [20]. Samples were sectioned to 10 μm vertically using a rotary microtome (Leica RM2255, Leica, Germany), dewaxed, rehydrated, stained with 1% safranin, counter-stained with 1% fast green, cleaned, and then fixed with neutral balata. Sections were imaged by positive fluorescence microscope (Leica DM6B, Leica, Germany).

### 2.8. Measurement of Watermelon Scion Dry Weight, Leaf Area of Watermelon Scion, and Rootstock Regrowth

Plants were sampled at day 14 after grafting. The fresh samples of watermelon scion were placed into a forced air oven at 105 °C for 15 min and then at 70 °C for 3 days to determine their dry weights. Leaf area of watermelon scion and rootstock regrowth was measured using an area meter LI-3100C (Li-Cor, Inc., Lincoln, NE, USA). Only the regrown rootstock plants were investigated and calculated for the regrown rootstock leaf area measurement.

### 2.9. Working Efficiency Measurement

The experiment was conducted by three skilled master students. The time used to physically remove the rootstock cotyledon epidermis in the new grafting method was counted, and the time used to manually remove the regrown rootstock 3 times in the traditional method was also counted. The measurement was conducted 3 times independently.

### 2.10. Data Analyses

All data were analyzed by Student's t-test using SPSS 25.0 software (SPSS Inc., Chicago, IL, USA). The figures were made using GraphPad Prism 8.0 (GraphPad Software Inc., San Diego, CA, USA). Significance between new grafting and traditional grafting method was set at $p < 0.05$ *, $p < 0.01$ ** or $p < 0.001$ ***, respectively.

## 3. Results

### 3.1. New Grafting Method Had no Significant Effect on the Graft Survival Rate

There was no significant difference on the graft survival rate between the new grafting method and the traditional grafting method for pumpkin and bottle gourd rootstocks, for both the hole insertion and one cotyledon grafting methods (Figure 3a–d). The graft survival rate of plants grafted onto pumpkin was 84%–98% (Figure 3a,b), while the value was 93%–100% for the plants grafted onto bottle gourd rootstock (Figure 3c,d).

### 3.2. New Grafting Method Largely Decreased the Rootstock Regrowth Rate

Compared with traditional grafting, few regrown rootstocks were seen in the plants using the new method (Figure 4a–d). The rootstock regrowth rate was decreased from 100% to 8% in pumpkin hole insertion using the new method as compared with traditional methods, and the value was decreased from 100% to 2% in one cotyledon grafting (Figure 5a). For bottle gourd rootstock, the rootstock regrowth rate was decreased from 100% to 23% using hole insertion and from 100% to 9% using one cotyledon grafting (Figure 5b). Anatomical observation of the pumpkin rootstock cotyledon base also showed that the new method inhibited the rootstock regrowth at earlier stages (24 and 36 h) after grafting (Figure 6a,b). Among the few regrown rootstock plants using the new method, the regrown leaf area was significantly smaller than the traditional one cotyledon grafting method, and no significant difference was observed on the hole insertion grafting (Figure 5c,d).

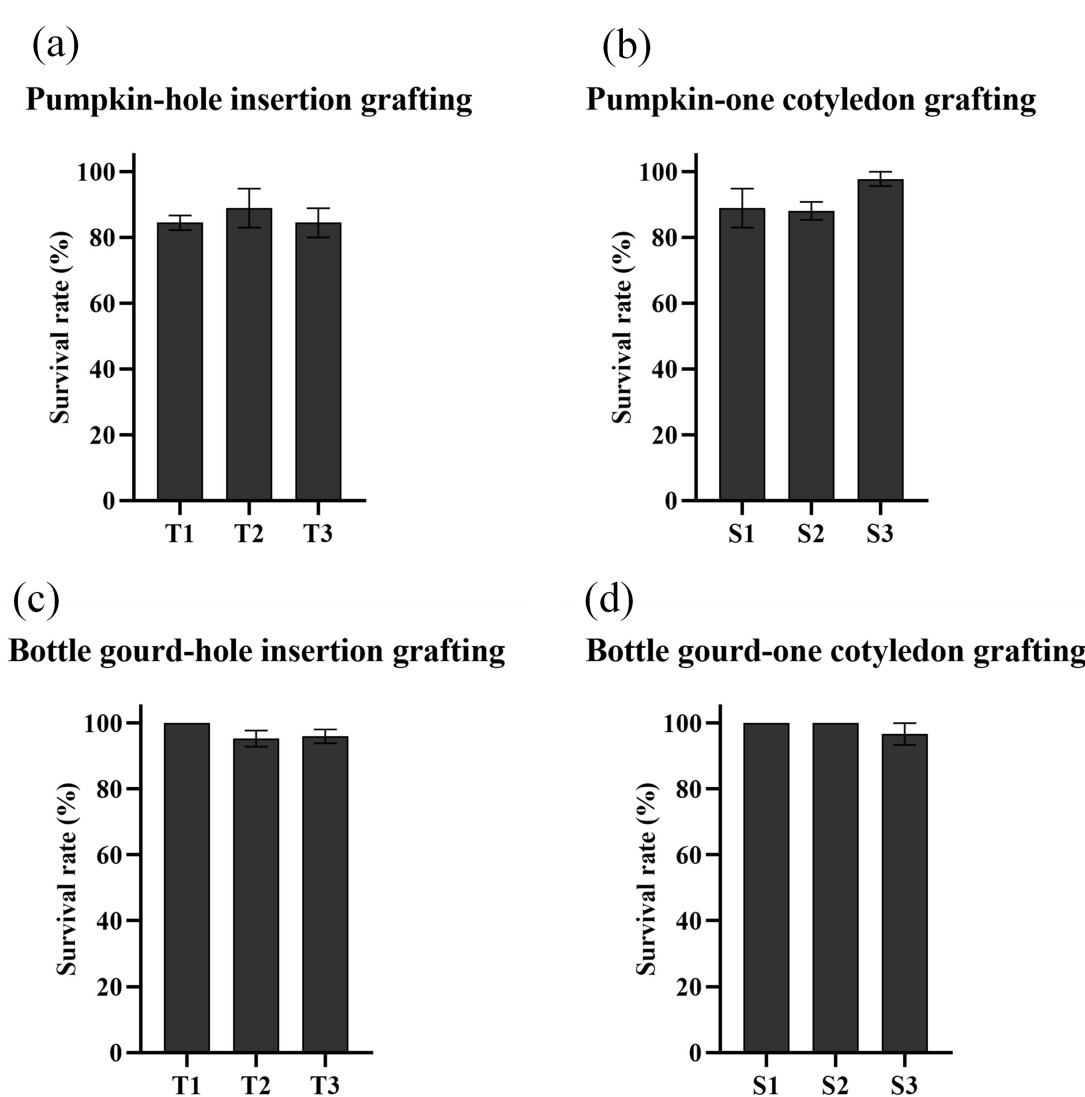

**Figure 3.** Graft survival rate at day 14 after grafting (*n* = 3). T1 and S1, new "hole insertion grafting" (**a**,**c**) and "one cotyledon grafting" (**b**,**d**), respectively, without the regrown rootstock removal; T2 and S2, traditional "hole insertion grafting" (**a**,**c**) and "one cotyledon grafting" (**b**,**d**), respectively, with manual regrown rootstock removal 3 times; T3 and S3, traditional "hole insertion grafting" (**a**,**c**) and "one cotyledon grafting" (**b**,**d**), respectively, without the regrown rootstock removal.

Compared with pumpkin traditional grafting without rootstock regrowth removal, the new grafting method significantly increased scion dry weights by 78% and 75% for hole insertion and one cotyledon grafting, respectively (Figure 7a,b). In the traditional hole insertion and one cotyledon grafting methods, removal of pumpkin regrown rootstock significantly increased scion dry weight compared with the traditional method without removal of regrown rootstock (Figure 7a,b). For bottle gourd rootstock, the new grafting method significantly increased scion dry weights by 25% and 24% for hole insertion and one cotyledon grafting, respectively, as compared with traditional grafting with regrown rootstock removal (Figure 7c,d).

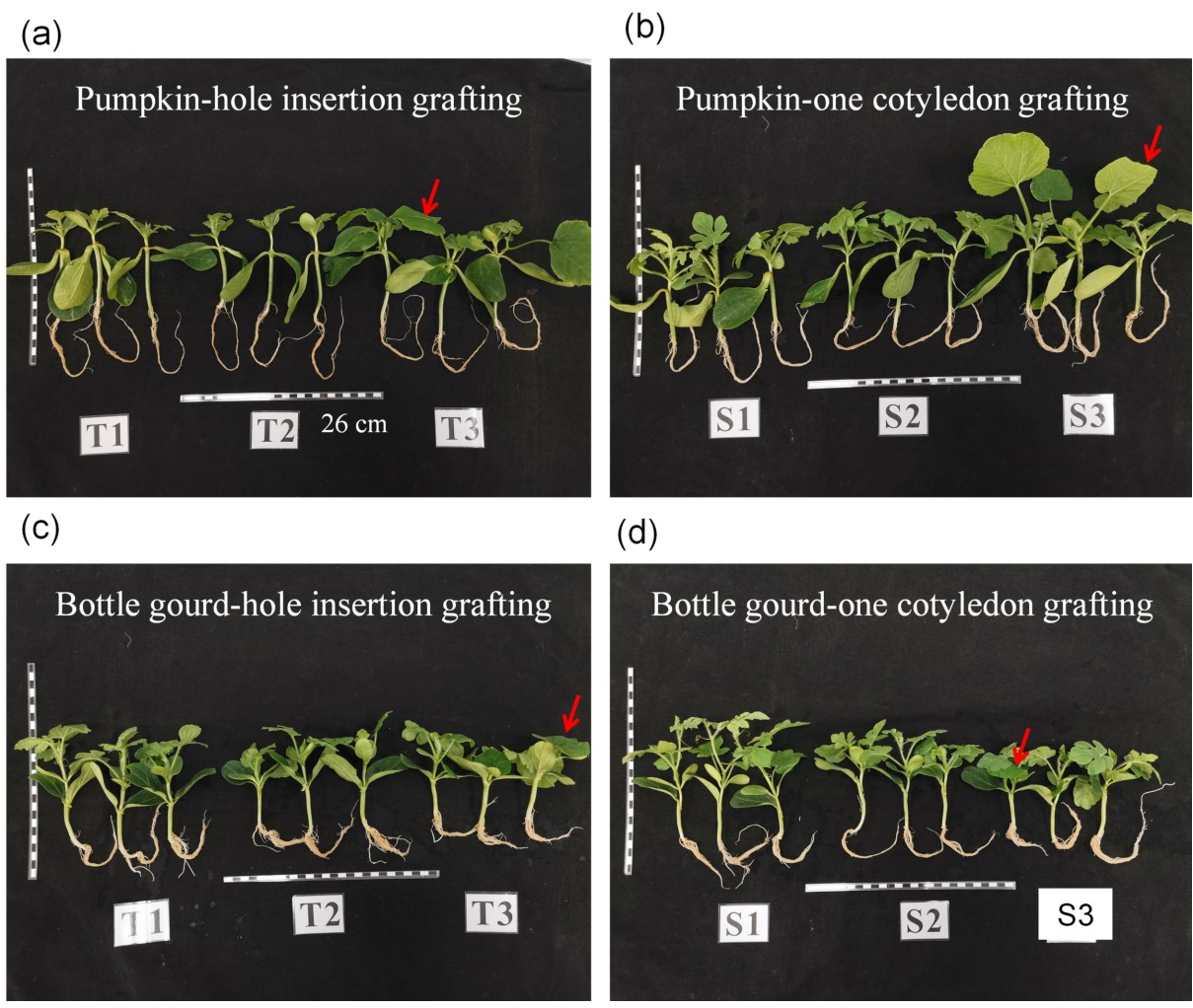

**Figure 4.** Watermelon growth pictures at day 14 after grafting. T1 and S1, new "hole insertion grafting" (**a**,**c**) and "one cotyledon grafting" (**b**,**d**), respectively, without the regrown rootstock removal; T2 and S2, traditional "hole insertion grafting" (**a**,**c**) and "one cotyledon grafting" (**b**,**d**), respectively, with manual regrown rootstock removal 3 times; T3 and S3, traditional "hole insertion grafting" (**a**,**c**) and "one cotyledon grafting" (**b**,**d**), respectively, without the regrown rootstock removal. Red arrow indicates rootstock regrowth.

### 3.3. New Grafting Method Enhances Scion Growth

Compared with traditional grafting without regrown rootstock removal, the new grafting method significantly increased scion leaf areas for both rootstocks by 49% and 67% for pumpkin hole insertion and one cotyledon grafting, respectively (Figure 8a,b), while the values were 26% and 25%, respectively for bottle gourd (Figure 8c,d). Removal of pumpkin rootstock regrowth was beneficial for the scion leaf area using the traditional method (Figure 7a,b).

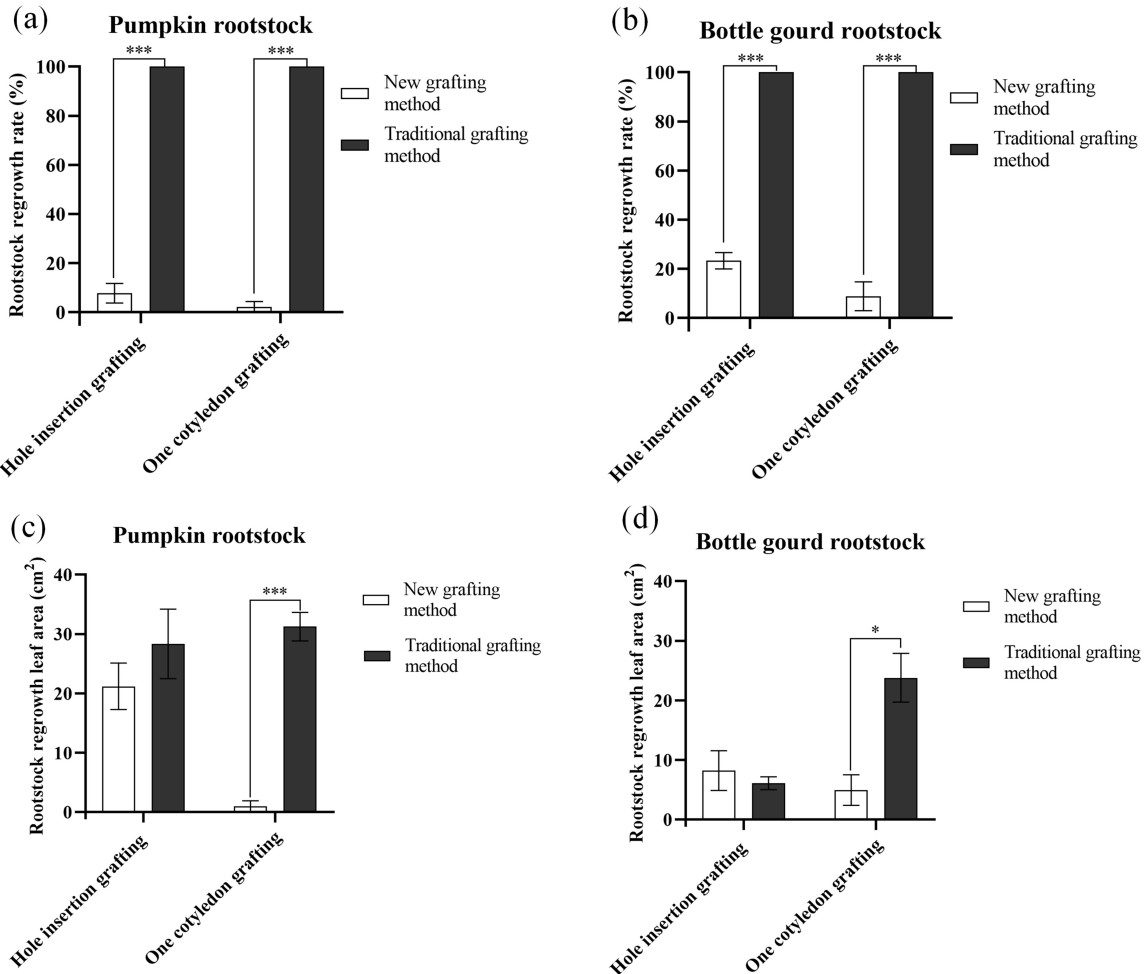

**Figure 5.** Rootstock regrowth rate (**a**,**b**) and regrown rootstock leaf area (**c**,**d**) at day 14 after grafting (*n* = 3). The rootstock regrowth was not removed for the new or the traditional grafting methods. For the measurement of regrown rootstock leaf area, only the regrown rootstock plants were investigated and calculated. * and *** indicate significant differences at *p* < 0.05 and 0.001 levels, respectively.

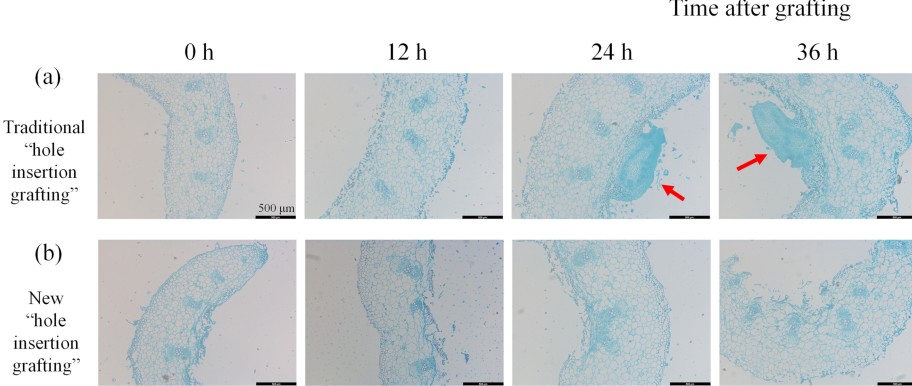

**Figure 6.** Anatomical observation of the pumpkin rootstock cotyledon base in the traditional (**a**) and the new "hole insertion grafting" (**b**) plants. Red arrow indicates rootstock regrowth.

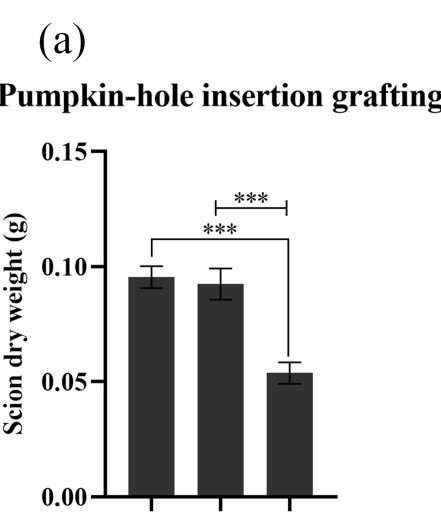

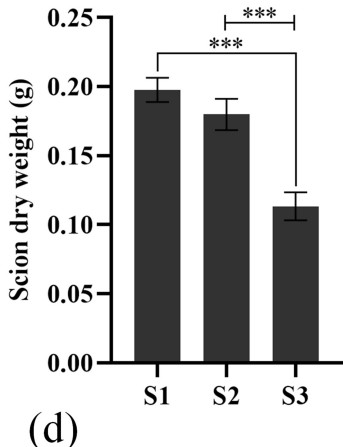

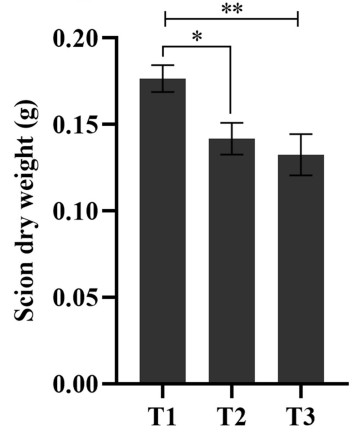

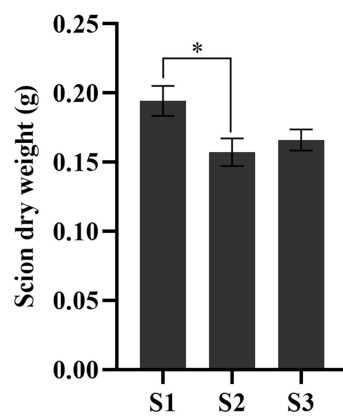

**Figure 7.** Watermelon scion dry weight at day 14 after grafting (*n* = 3). T1 and S1, new "hole insertion grafting" (**a**,**c**) and "one cotyledon grafting" (**b**,**d**), respectively, without regrown rootstock removal; T2 and S2, traditional "hole insertion grafting" (**a**,**c**) and "one cotyledon grafting" (**b**,**d**), respectively, with manual regrown rootstock removal 3 times; T3 and S3, traditional "hole insertion grafting" (**a**,**c**) and "one cotyledon grafting" (**b**,**d**), respectively, without regrown rootstock removal. *, **, and *** indicate significant differences at $p < 0.05$, 0.01, and 0.001 levels.

### 3.4. New Grafting Method Is Labor Saving

Compared with traditional grafting, the new grafting method increased the time to physically remove the epidermis of rootstock cotyledon but decreased the time used for the removal of rootstock regrowth (Figure 9). By calculation of the working efficiency, the time used for the new method is much shorter than the traditional method: for hole insertion grafting, 4.2 s/plant vs. 9.3 s/plant; for one cotyledon grafting, 4.2 s/plant vs. 8.8 s/plant. Considering very few regrown rootstocks appeared in the new grafting method (Figure 9), the amount of labor required to remove the rootstock regrowth was very small, thus, this new grafting method is labor-saving compared with traditional grafting methods.

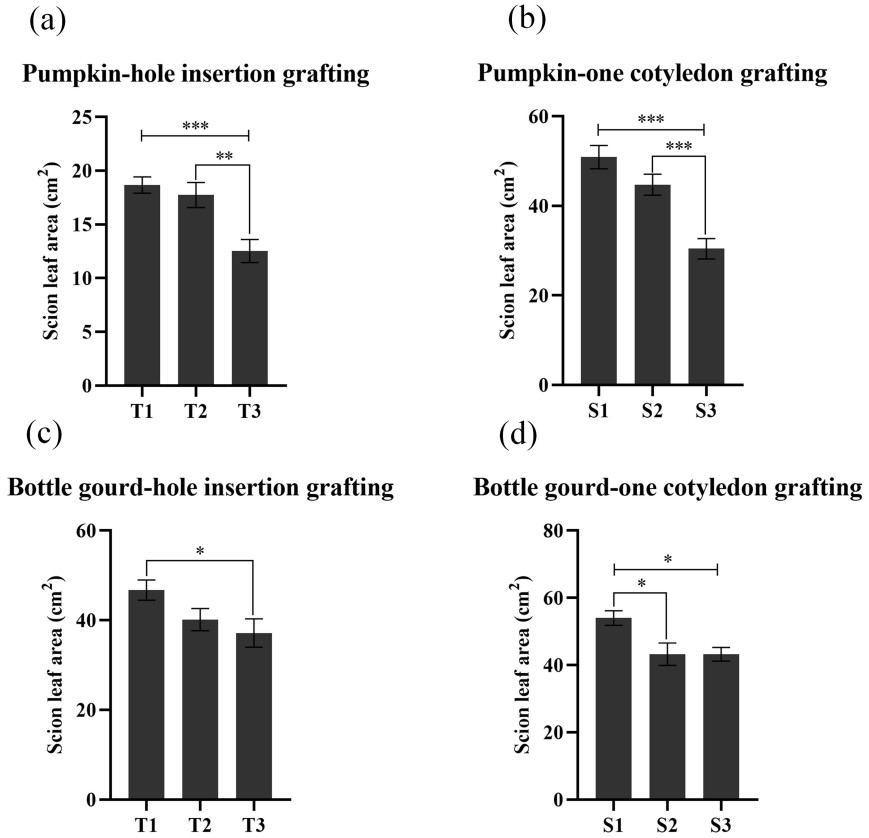

**Figure 8.** Watermelon scion leaf area at day 14 after grafting (*n* = 3). T1 and S1, new "hole insertion grafting" (**a**,**c**) and "one cotyledon grafting" (**b**,**d**), respectively, without regrown rootstock removal; T2 and S2, traditional "hole insertion grafting" (**a**,**c**) and "one cotyledon grafting" (**b**,**d**), respectively, with manual regrown rootstock removal 3 times; T3 and S3, traditional "hole insertion grafting" (**a**,**c**) and "one cotyledon grafting" (**b**,**d**), respectively, without regrown rootstock removal. *, **, and *** indicate significant differences at $p < 0.05$, 0.01, and 0.001 levels.

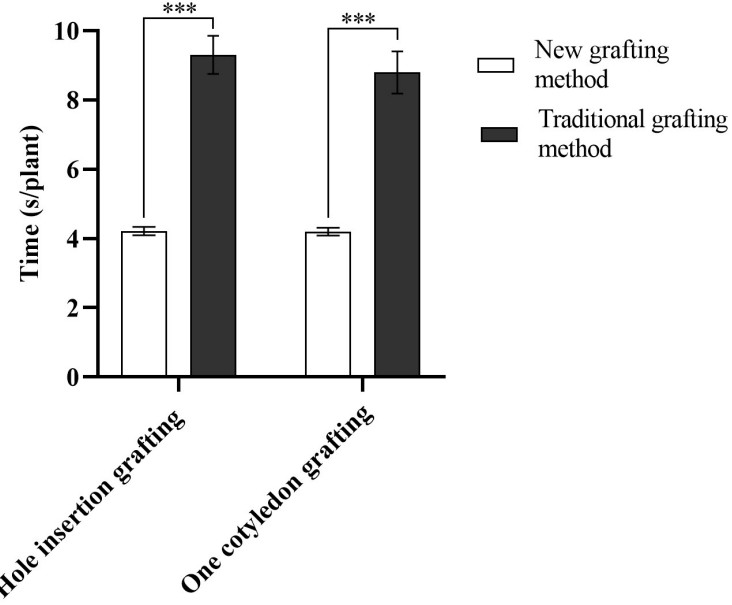

**Figure 9.** Time used to physically remove the pumpkin rootstock cotyledon epidermis for the new grafting method and the time required to remove rootstock regrowth for the traditional grafting method (*n* = 3). *** indicates significant difference at $p < 0.001$ level.

## 4. Discussion

Grafting of watermelon scions onto pumpkin or bottle gourd rootstocks has been practiced in many of the major watermelon production regions of the world [2,5]. However, grafting needs additional labor, such as the removal of rootstock regrowth. Thus, labor-efficient grafting methods have been recognized as a key to success in production of grafted watermelon seedlings on a large scale [5]. Rootstock regrowth is a major concern for watermelons that are grafted with at least one intact rootstock cotyledon, as it is difficult to remove all the rootstock bud meristem tissue [5]. Many commercial rootstocks are vigorous and quickly overtake the scion variety if allowed to grow, thus scouting for and removal of rootstock regrowth is required. In fact, only a small number of studies focused on the rootstock regrowth topic, although it is a big problem in practice. There was no rootstock regrowth by splice grafting; however, the graft survival rate was only 18% at day 21 after grafting [15]. Exogenous treatments of sucrose with antitranspirant to rootstock seedlings before grafting can increase the survival of splice grafted watermelon to 90%, however, graft healing environment should be precisely controlled, and more labor is needed for the applications of chemicals [15–17]. In addition, application of fatty alcohol compounds to watermelon rootstock meristems can decrease the rootstock regrowth rate to 5%, however, the concentration of fatty alcohol must be carefully selected, as higher concentrations of fatty alcohol can damage more than 90% of plants [18]. Rootstock regrowth can also be controlled by genetic approaches. Li et al. [21] demonstrated that expression of both iaaM (tryptophan-2-monooxygenase gene, an auxin biosynthesis gene) and CKX (a cytokinin degradation gene) genes predominantly in roots of tobacco rootstock can inhibit lateral bud release from rootstock.

In this study, considering the high graft survival rate (>84% in pumpkin rootstock, 100% in bottle gourd rootstock) and scion growth, the low regrowth rate of rootstock, and the fact that there was no obvious plant damage, the new grafting method has significant advantages compared with traditional grafting (Figures 3–9) and the methods described in previous studies [15–18]. Compared with traditional hole insertion and one cotyledon grafting methods, although the new method requires the procedure of physically removing the epidermis of the rootstock cotyledon base, the procedure of removing the rootstock regrowth is not needed. It should be indicated that rootstock regrowth removal is often conducted three times at the seedling stage. There was increased scion growth using the new grafting method, which could be attributed to the enhanced distribution of photosynthate to the scion rather than the absent rootstock regrowth [15,16,18]. Compared with traditional grafting methods, the time used to produce one grafted seedling could be decreased by about 5 s (Figure 9). Obviously, labor intensive grafting propagation potentially could benefit from the effective use of this grafting method which minimizes labor inputs, thus reducing the overall cost of producing grafted watermelon seedlings and helping increase the adoption of grafted cucurbit plants in the world.

## 5. Conclusions

This study developed a new grafting method; the epidermis of the rootstock cotyledon base was physically removed by a new grafting tool. Using this new method, the rootstock regrowth rate was decreased from 100% to less than 23%, and watermelon scion dry weight was increased by more than 17% compared with traditional grafting. In addition, the new method can save about 5 s in producing one grafted watermelon seedling. Considering seedling growth and labor input, this new method is cost-effective and has great potential application prospects in practice.

## 6. Patents

Yuan Huang, Changjin Liu, Zhilong Bie, Weiguo Lin, Xiangshuai Wu, Xiaohu Fu. A new cucurbit grafting tool used to inhibit rootstock regrowth and a new grafting method, Application Number: 202110518435.5, China National Intellectual Property Administration.

**Supplementary Materials:** The following are available online at https://www.mdpi.com/article/10 .3390/agriculture11090812/s1, Supplementary Video S1: Method to show how the new grafting tool is used, https://www.preprints.org/manuscript/202107.0693/v1/download/supplementary.

**Author Contributions:** Conceptualization, Y.H. and Z.B.; methodology, C.L.; formal analysis, C.L. and X.W.; investigation, C.L., X.W. and M.X.; new grafting tool development, Y.H., W.L. and X.F.; pictures drawing, C.F.; writing—original draft preparation, Y.H. and C.L.; writing—review and editing, Y.H.; supervision, Y.H.; project administration, Y.H.; funding acquisition, Y.H. and Z.B. All authors have read and agreed to the published version of the manuscript.

**Funding:** This research was funded by the National Key Research and Development Program of China (2019YFD1001900), National Natural Science Foundation of China (31972434), China Agriculture Research System of MOF and MORA (CARS-25), Hubei Provincial Natural Science Foundation of China (2019CFA017), and Huazhong Agricultural University-Agricultural Genomics Institute at Shenzhen, Chinese Academy of Agricultural Sciences Cooperation Fund (SZYJY2021005).

**Institutional Review Board Statement:** Not applicable.

**Informed Consent Statement:** Not applicable.

**Data Availability Statement:** The data presented in this study are available on request from the corresponding author.

**Acknowledgments:** The authors are thankful to Xiaoyang Wei for revising the English text.

**Conflicts of Interest:** The authors declare no conflict of interest. The funders had no role in the design of the study; in the collection, analyses, or interpretation of data; in the writing of the manuscript, or in the decision to publish the results.

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
