# Peer review of "A New Grafting Method for Watermelon to Inhibit Rootstock Regrowth and Enhance Scion Growth"

_agriculture, doi:10.3390/agriculture11090812_

Round 1

Reviewer 1 Report

This manuscript presents an interesting new approach in watermelon grafting to reduce rootstock regrowth and therefore benefiting both scion growth and plant management. However, calling it a "new grafting method" might be confusing, since the basis of this study is using a new grafting tool to improve previous grafting methods like "one cotyledon grafting" and "hole insertion grafting". Anyhow, the paper contributes to the developement of new protocols that might be clearly useful in watermelon production.

Discussion can be improved, specially from line 244. Other hormones and processes involved in axillary bud regeneration should be mentioned. Though, the hypothesis established in line 254 is define as speculative, it would highly benefit from some reference, if possible.

Changes to the language should be made.

line 49 - "problem, new grafting method was developed"

line 52 - "chemical compunds [...] was also used"

line 59 - It should be rephrased for an easier understanding

line 182 - "roottsock"

line 235 - It should be rephrased for an easier understanding

Author Response

Thank you very much for your comments, please see the attached file.

Reviewer 2 Report

The subject of article is interesting for practice.

The manuscript is general written in a clear way, easy to read and with appropriate reference to data in figures. Authors should add more information in introduction about plant cultivation and sign how much this problem is important in practice.

The material and method needs improvement: conditions of experiment (protection of plants eg. used fertilization and plant protection products) and make more attitude for abbreviation, where in many places is not understandable (eg. T1, S1).

In my opinion conclusion is not precise and needs correction.

All suggestions are included in the manuscript.

Author Response

(The authors gave the same response as above.)

Reviewer 3 Report

The manuscript by Changjin Liu et al. provides a study of „A new grafting method for watermelon to inhibit rootstock re-growth and enhance scion growth“. This study presents a new grafting method for watermelon to inhibit rootstock regrowth and enhance scion growth, which is cost-effective for grafted watermelon seedlings. The motivation for this work is clear – new grafting method development. The author presents a very extensive work, and the results are very well presented. The supplementary video is also useful. I have a few comments below and recommendations.

  1. Section – 2. Materials and Methods - Subsections are unnecessary (2.1 – 2.10). It is confusing.
  2. Page 3 – Figure 1 is too large.
  3. Line 98 (line 104) – Figure 2 or Figures 2a and 2b??
  4. Figures 3,5,7 and 8 are too small. Change these figures, please. I think that the size of Figure 9 is good.

Author Response

(The authors gave the same response as above.)

Reviewer 4 Report

Here are some comments and suggestions:

Abstract

The abstract does not reflect the content and the findings of this study.

In particular:

Line 16: use other phrases than ‘’ biotic and abiotic stressors’’ as it reflects that includes some stress treatment in your work that is not the case.

Line 17-18: The whole sentence needs major revision as it is strangely formulated and do not use the same sentences that are written in elsewhere in the manuscript (duplicated from Line 41-42).

Line: 19: Rather use ‘’physically removed’’ than ‘’ destroy (remove)’’. Try to put the Latin names for all of the plants that are used when they appear first (pumpkin (Cucurbita pepo) and bottle gourd (Lagrenaria siceraria)).

Line 21. ‘’(2-23%)’’ is not clearly reflecting the results. Try to open it and give a more detail sentences for your results in the abstract.

Such as for line 20: ‘’Compared with the traditional grafting, the new grafting method had significantly reduced the pumpkin rootstock regrowth rate from 100% to 8% and 2% in hole insertion and one cotyledon grafting, respectively. These attenuated rates for bottle gourd rootstock regrowth were 23% and 9% in hole insertion and one cotyledon grafting, respectively.’’

Line 21: ‘’ higher watermelon scion dry weight and leaf area’’. Put this in a separate sentence and try to give some values for specificity.

Line 22-25: I suggest reformulating this sentence. In the current form, it is difficult to comprehend the message. Maybe if you divide it to two sentences separating the hole insertion and one cotyledon grafting it would be more clear.

Line 25-27: needs a vigorous revision. Despite the primitive English that has to be transformed to a standard level, try to formulate a ‘’selling text’’ here, as this is the most important section of your abstract.

Introduction

The content is adequate but the sentences can be better formulated.

Line 31: change ‘’ …scion phenotypes, such as soil borne disease resistance and fruit yield traits’’ to ‘’ ... scions for disease resistance and enhanced yield traits’’

Line 58: I think you do not need to summarize the results of your work here.

Materials and methods

Line 66: I’m not sure if ‘’ … without rootstock regrowth removal’’ is a correct phrase. Try to come up with a better phrase.

Line 67: Use another term/s for ‘’treatment’’ or clearly write what do you mean by treatment/s.

Line 76: ‘’day’’ or ‘’day and night’’ relative humidity was 65-85%, respectively?

Line 79-81: better to give morphological indications than the day’s number. Such as days after the appearance of cotyledons or the size of the first leaves in cm in diameter.

Line 83: ‘’light’’ to ‘’light intensity’’.

Line 84: ‘’was kept normal’’ to “increased to”

Line 115: change “the rootstock regrowth” to “ the regrown rootstock”.

Line 118: is it necessary to have “earlier” here?

Line 119: remove extra spaces between the words.

Line 120: Shouldn’t it be specified for “longitudinal sectioning” instead of “Paraffin section”? Give some brief details if you can.

Line 131-133: Give a better explanation. I guess the intended meaning is not clear here: “… rootstock regrowth 3 times in the traditional method was counted”.

Line 134: More details such as the methods for validation of the homogeneity of the variances are welcome.

Results

Line 147: I advise to give a full and clear description for T1-3 and S1-3 in a table. It is a bit difficult to follow the codes in the graph caption.

Line 160: Change “regrowth” to “regrown”.

Line 161: Change “lower” to “smaller”.

Line 165: Modify as suggested above for “rootstock regrowth removal”.

Line 169: Charts can be presented in larger size and better resolution.

Line 178: Modify “rootstock regrowth removal” everywhere!

Line 179: Remove the 2nd “increased”.

Line 180-181: Rephrase and complete the whole sentence to deliver the intended meaning.

Line 183: Remove the “increased”.

Line 193: Remove the 2nd “increased”.

Line 195-196: Rephrase and complete the whole sentence to deliver the intended meaning.

Line 169 and 199: Comparing the leaf sizes in the images at 14th day, it is odd that the leaf area of each individual rootstock plant can be as large as around 80 or 100 cm2. The same for scion plants leaf area of up to 150 cm2 .Please clarify!

Line 213: The chart size can be smaller.

Discussion

The discussion section of this manuscript needs a major revision. All of the context that is already stated in the introduction should be removed and only the obtained results should be discussed with the other similarly approached studies.

Line 218: Sub numbering is not necessary. You can combine the two and make a compact and strait forward discussion.

Line 225-229: Already mentioned in the introduction part.

Line 230: Change the “labor output”.

Line 238: What is “4 s”?

Line 246-257: This part in its current form is very out of context. You should make a proper discussion. The linkage between the CK, its content, signaling and the physical removal of cells need a much deeper study and I suggest not to get to that direction as the message of this paper is a practical solution for horticulture.

You must compare your results with the other innovative solutions published on controlling the rootstock re-growth by means of some % values. That would include at least 6-7 more articles to your references.

Conclusion

You may exclude that at all, or make a compact summary including the main criteria of your work. The important elements, the outcomes and the potential application of your method.

Line 266: The schematic presentation is a good idea but may be it is better to have it in supplementary materials. As the content is mainly presented in Fig.2.  Try to give more space between the lines, the cartoons and the texts and rearrange the areas to have ‘’a’’ and ‘’b’’ and then ‘’c’’ and ‘’d’’ in the figure title.

Author Response

(The authors gave the same response as above.)

Reviewer 5 Report

This is interesting article about a new grafting method for watermelon to inhibit rootstock regrowth and enhance scion growth.

It is a well written paper and presents detailed results of a comprehensive study examining the potential benefit of a new grafting method, in which the epidermis of rootstock cotyledon base was removed by a new grafting tool. The experiments were conducted with adequate replication and well-organized experimental design. The work is important for the research field and is being well conducted. I have only some small suggestions:

Lines 62-67. It is not clear to me whether this experiment was carried out in just one year or whether you tested this new grafting method for several years, for the results which you have reached. Please specify this.

Line 251. I think the term WUS refers to the homeodomain transcription factor WUSCHEL, but I suggest you specify this for readers.

Line 262. In my opinion, Figure 10 should not be to the Conclusions section. My suggestion is to moving that figure to the Results section.

Author Response

(The authors gave the same response as above.)

Round 2

Reviewer 4 Report

The manuscript is significantly improved but still a comprehensive English language editing is required. I advise you to ask a native English speaker to read and revise the applied language.
Line 20: remove the “,” after “(Lagenaria siceraria,)”
There are several sentences that have to be corrected such as for example:
Line 44: “Grafting is a commonly used technique for watermelon” to “Grafting is a commonly used technique in watermelon production”.

Author Response

  1. The manuscript is significantly improved but still a comprehensive English language editing is required. I advise you to ask a native English speaker to read and revise the applied language.

Response: The MS has been revised by my colleague Prof. Robert M. Larkin.

  1. Line 20: remove the “,” after “(Lagenaria siceraria,)”

Response: Revised.

  1. Line 44: “Grafting is a commonly used technique for watermelon” to “Grafting is a commonly used technique in watermelon production”.

Response: Revised.

  1. Other revisions, please see the red text of revised MS.
